# 8-Isoprostanes and Asymmetric Dimethylarginine as Predictors of Mortality in Patients Following Coronary Bypass Surgery: A Long-Term Follow-Up Study

**DOI:** 10.3390/jcm11010246

**Published:** 2022-01-04

**Authors:** Aleksandra Gołąb, Dariusz Plicner, Anna Rzucidło-Hymczak, Lidia Tomkiewicz-Pająk, Bogusław Gawęda, Bogusław Kapelak, Anetta Undas

**Affiliations:** 1Faculty of Medicine and Dentistry, Pomeranian Medical University, 70-204 Szczecin, Poland; olagoab13@gmail.com; 2Unit of Experimental Cardiology and Cardiac Surgery, Faculty of Medicine and Health Sciences, Andrzej Frycz Modrzewski Krakow University, 30-705 Krakow, Poland; 3Department of Cardiovascular Surgery and Transplantation, John Paul II Hospital, 31-202 Krakow, Poland; bogus.kapelak@gmail.com; 4Department of Pediatric Infectious Diseases and Pediatric Hepatology, John Paul II Hospital, 31-202 Krakow, Poland; hymczak@op.pl; 5Department of Cardiac and Vascular Diseases, John Paul II Hospital, 31-202 Krakow, Poland; ltom@wp.pl; 6Institute of Cardiology, Jagiellonian University Medical College, 31-008 Krakow, Poland; mmundas@cyf-kr.edu.pl; 7Division of Cardiovascular Surgery, St. Jadwiga Provincial Clinical Hospital, Medical College of Rzeszow University, 35-310 Rzeszow, Poland; bgaweda@onet.eu; 8Center for Research and Innovative Technology, John Paul II Hospital, 31-202 Krakow, Poland

**Keywords:** coronary artery bypass grafting, asymmetric dimethylarginine, oxidative stress, 8-iso-prostaglandin F_2α_, long-term cardiovascular death

## Abstract

Background: We previously demonstrated that enhanced oxidative stress and reduced nitric oxide bioavailability are associated with unfavorable outcomes early after coronary artery bypass grafting. It is not known whether these processes may impact long-term results. We sought to assess whether during long-term follow-up, markers of oxidative stress and nitric oxide bioavailability may predict cardiovascular mortality following bypass surgery. Methods: We studied 152 consecutive patients (118 men, age 65.2 ± 8.3 years) who underwent elective, primary, isolated on-pump bypass surgery. We measured plasma 8-iso-prostaglandin F2α and asymmetric dimethylarginine before surgery and twice after surgery (18–36 h and 5–7 days). We assessed all-cause and cardiovascular death in relation to these two biomarkers during a mean follow-up time of 11.7 years. Results: The overall mortality was 44.7% (4.7 per 100 patient-years) and cardiovascular mortality was 21.0% (2.2 per 100 patient-years). Baseline 8-iso-prostaglandin F2α was associated with cardiovascular mortality (HR 1 pg/mL 1.010, 95% CI 1.001–1.021, *p* = 0.036) with the optimal cut-off ≤ 364 pg/mL for higher survival rate (HR 0.460, 95% CI 0.224–0.942, *p* = 0.030). Asymmetric dimethylarginine > 1.01 μmol/L measured 18–36 h after surgery also predicted cardiovascular death (HR 2.467, 95% CI 1.140–5.340, *p* = 0.020). Additionally, elevated 8-iso-prostaglandin F2α measured at the same time point associated with all-cause mortality (HR 1 pg/mL 1.007, 95% CI 1.000–1.014, *p* = 0.048). Conclusions: Our findings indicate that in advanced coronary disease, increased oxidative stress, reflected by 8-iso-prostaglandin F2α before bypass surgery and enhanced asymmetric dimethylarginine accumulation just after the surgery are associated with cardiovascular death during long-term follow-up

## 1. Introduction

The disproportion between the production of reactive oxygen species (ROS) and antioxidant clearance, termed as oxidative stress, has been involved in the pathogenesis of several disorders [1,2,3]. Numerous reports have demonstrated that enhanced oxidative stress is associated with coronary artery disease [4]. Several cardiovascular risk factors such as diabetes, hypertension, dyslipidemia, smoking, and obesity are related to an increased production of ROS [4,5]. Isoprostanes, isomers of prostaglandins derived from non-enzymatic ROS-induced lipid peroxidation under oxidative stress, contribute to oxidative injury by modifying platelet function and by reducing the antiplatelet activity of nitric oxide, which can contribute to the progression of atherosclerosis [6]. A specific class of isoprostanes, the 8-iso-prostaglandin F_2α_ (8-iso-PGF_2α_), is considered a useful marker for assessing endogenous oxidative stress in clinical studies, because isoprostanes are stable and can be determined in all biological fluids [5]. Asymmetric dimethylarginine (ADMA), a natural inhibitor of nitric oxide synthase, has been shown to accumulate in oxidative stress thus reducing the nitric oxide bioavailability which, among others, results in the inhibition of ROS inactivation [7].

We have shown that a significant increase of ADMA and oxidative stress, reflected by 8-iso-PGF_2α_ is detected within the first days after coronary artery bypass grafting (CABG) and is associated with unfavorable early post-surgery outcomes, including postoperative myocardial infarction and in-hospital cardiovascular death [8]. It is unclear whether these parameters could be of value in predicting mortality during long-term follow-up. Some reports have examined the predictive role of oxidative stress level for cardiovascular events rate, but to the best of our knowledge, there have been no studies investigating oxidative stress markers and ADMA during long-term follow-up after CABG surgery [4,9,10]. In the present study we tested the hypothesis that these two biomarkers may predict cardiovascular mortality following bypass surgery.

## 2. Materials and Methods

### 2.1. Patients

The study population comprised 152 consecutive patients undergoing primary, elective and isolated CABG surgery, with the use of cardiopulmonary bypass. The study group was described in detail previously [8]. Briefly, all CABG procedures were performed between January and March 2008 and each patient received both an internal mammary artery and a saphenous vein grafts. Indications for bypass surgery were consistent with the American College of Cardiology and American Heart Association guidelines for CABG [11]. The exclusion criteria were a history of any cardiac surgery or percutaneous coronary intervention within 1 month before enrolment, renal failure and acute coronary syndromes or unstable hemodynamic conditions on admission. The original report from 2014 showed the results of the prospective study and this current study represents a post hoc analysis [8]. The study was performed in accordance with the Declaration of Helsinki and the protocol was accepted by the Andrzej Frycz Modrzewski Krakow University Ethical Committee (KBKA/14/O/2021), which waived the need for informed approval because of the retrospective type of analysis. 

### 2.2. Laboratory Investigations

We measured plasma 8-iso-PGF_2α_ and ADMA concentrations before and twice after surgery (18–36 h and 5–7 days). Plasma 8-iso-PGF_2α_ was measured using a commercially accessible enzyme-linked immunosorbent assay (8-Isoprostane ELISA Kit, Cayman Chemical, MI, USA). Plasma ADMA concentration was determined by isocratic high-performance liquid chromatography, as described [12]. Standard laboratory parameters were assessed using routine techniques. All variables were evaluated in duplicate by a researcher blinded to the sample origin. All intra-assay and inter-assay coefficients of variation were below 7%.

### 2.3. Follow-Up

We recorded all-cause and cardiovascular deaths which occurred until 31 October 2019. The vital status and eventually the cause and date of death was collected from the National Mortality Registry maintained by the State Systems Department of Ministry of Digital Affairs in Poland. The reasons of mortality were encrypted in accordance with the 10th Revision of International Classification of Diseases (ICD-10). The main endpoints were death due to heart diseases (ICD-10 codes I10-I52), cerebrovascular diseases (ICD-10 codes I60-I69), other diseases of the circulatory system (ICD-10 codes I70-I99), malignant neoplasms (ICD-10 codes C00-C97), respiratory diseases (ICD-10 codes J00-J99), accidents (ICD-10 codes V01-X59), diabetes (ICD-10 codes E10-E14) and other causes (rest of the ICD-10 codes). ICD codes I10-I52, I60-I69 and I70-I99 were considered to contribute to the cardiovascular death. All-cause mortality was defined as death due to any causes.

### 2.4. Statistical Analysis

Categorical variables are presented as numbers and percentages. Continuous variables are expressed as mean ± standard deviation. Normality was assessed by Shapiro–Wilk test. Equality of variances was assessed using Levene’s test. Differences between groups were compared using the Student’s or Welch’s *t*-test depending on the equality of variances for normally distributed variables. The Mann–Whitney U-test was used for non-normally distributed continuous variables. Ordinal variables were compared using the Cochran–Armitage test for trend. Nominal variables were compared by the Pearson’s chi-squared test or Fisher’s exact test as appropriate.

To analyze event-free survival in selected risk groups, Kaplan–Meier curves were drawn. The log-rank test was used to test the differences in the outcomes between the groups. Determinants of late total death and cardiovascular death were determined by univariable and multivariable Cox regression models. All baseline parameters that were significantly associated with the outcome in univariable models were included in the multivariable logistic regression model to predict cardiovascular death. Receiver operating characteristic (ROC) curves were evaluated to determine the optimal cut-off values for variables. A p-value below 0.05 was considered statistically significant. Statistical analyses were performed with JMP^®^ (Version 14.2.0, SAS Institute INC., Cary, NC, USA) and using R (Version 3.4.1, R Core Team, Vienna, Austria).

## 3. Results

### 3.1. Demographic and Clinical Characteristics

As shown in Table 1, the mean baseline age of all enrolled patients was 65.2 ± 8.3 years and most were men (77.6%). The mean follow-up time was 11.7 ± 0.1 years. None of the patients was lost to follow-up. During the follow-up we documented 68 deaths, including 32 from cardiovascular diseases, which accounted for 47.1% of total deaths (Table 2). The overall mortality rate was 44.7% (4.7 per 100 patient-years) and cardiovascular mortality was 21.0% (2.2 per 100 patient-years).

Lower body mass index and type 2 diabetes were associated with higher overall mortality. Preoperative use of beta blockers was associated with longer overall survival, while statin treatment before surgery was a predictor of better cardiovascular survival. Additionally, remaining in intensive care unit 2 or more days was a risk factor for both types of long-term mortality (Table 1 and Table 3).

### 3.2. 8-iso-Prostaglandin F_2__α_

Cox hazards regression analysis (adjusted for age and sex) showed that a higher baseline level of 8-iso-PGF_2α_ was a risk factor for long-term cardiovascular mortality (hazard ratio [HR] 1 pg/mL 1.010, 95% confidence interval [CI] 1.001–1.021, *p* = 0.036). The ROC curve analysis showed the optimal cut-off of baseline 8-iso-PGF_2α_ to be ≤364 pg/mL for lower risk of long-term cardiovascular death (HR 0.460, 95% CI 0.224–0.942, *p* = 0.030). Kaplan–Meier curves revealed lower survival rate in patients with baseline 8-iso-PGF_2α_ > 364 pg/mL (log-rank test *p* = 0.039, Figure 1). Rise of 8-iso-PGF_2α_ between baseline and 18–36 h after the surgery > 135 pg/mL was also a risk factor for long-term cardiovascular death (HR 2.701, 95% CI 0.932–7.829, *p* = 0.041). A marked increase of oxidative stress detected within 18–36 h following CABG was similarly associated with all-cause long-term mortality (HR 1 pg/mL 1.007, 95% CI 1.000–1.014, *p* = 0.048). An 8-iso-PGF_2α_ concentration determined 5–7 days after CABG was not associated with mortality during follow-up (Table 4 and Table 5). 

Cut-offs were estimated based on the receiver operating characteristic curves. 8-iso-PGF_2α_: 8-iso-prostaglandin F_2α_; CABG: Coronary artery bypass grafting.

### 3.3. Asymmetric Dimethylarginine

Cox regression model (adjusted for age and sex) revealed that a ROC estimated ADMA > 1.01 μmol/L 18–36 h after CABG was a risk factor for long-term cardiovascular death (HR 2.467, 95% CI 1.140–5.340, *p* = 0.020). Kaplan–Meier curves showed lower survival rate in patients 18–36 h following surgery with ADMA > 1.01 μmol/L (log-rank test *p* = 0.034, Figure 2). Moreover, rise of ADMA > 0.44 μmol/L measured between baseline and 18–36 h following CABG was positively associated with long-term risk of cardiovascular death (HR 2.192, 95% CI 1.017–4.728, *p* = 0.047). Kaplan–Meier curves also showed lower probability of survival in patients with difference of ADMA concentration between baseline and 18–36 h after surgery > 0.44 μmol/L (log-rank test *p* = 0.041). Similarly to 8-iso-PGF_2α_, ADMA concentrations 5–7 days following surgery were not associated with morality during follow-up (Table 4 and Table 5). Additionally, there was a positive correlation between baseline 8-iso-PGF_2α_ and ADMA level 18–36 h after CABG (r = 0.81, *p* = 0.003). 

Cut-offs were estimated based on the receiver operating characteristic curves. ADMA: Asymmetric dimethylarginine; CABG: Coronary artery bypass grafting.

The multivariable model showed that following factors were associated with increased risk of cardiovascular death: age (HR per 10 years 1.827, 95% CI 1.138–3.021, *p* = 0.015), no preoperative statin treatment (HR 2.334, 95% CI 1.006–5,416, *p* = 0.048), baseline 8-iso-PGF_2α_ (HR 1 pg/mL 1.021, 95% CI 1.003–1.020, *p* = 0.043) and ADMA level > 1.01 μmol/L 18–36 h after CABG (HR 2.539, 95% CI 1.165–5.532, *p* = 0.019).

## 4. Discussion

In this study we showed that increased oxidative stress before CABG surgery and enhanced ADMA accumulation just after the procedure were associated with higher risk of cardiovascular death during an almost 12-year follow-up period. To the best of our knowledge, this report is the first to assess the value of 8-isoprostanes and plasma ADMA concentrations as long-term risk prediction in patients following CABG. We previously demonstrated that patients who suffered from postoperative myocardial infarction (or died of myocardial infarction) early after CABG had higher levels of these markers, both before and after surgery [8]. The present long-term follow-up study significantly extends the previous observations by demonstrating that persistently enhanced oxidative stress is predictive of mortality following CABG and its impact is still detectable after 12 years, which implies that it cannot be easily abolished or minimized over time in subjects with advanced coronary artery disease and larger burden of atherosclerosis. Long-term survival in our group of patients was found to be comparable to the data shown by other reports [13,14].

There is evidence that numerous well-established cardiovascular risk factors increase pro-oxidant compounds such as 8-iso-PGF_2α_ [4,5,15]. The current oxidative injury model of atherosclerosis suggests that several risk factors for atherosclerosis stimulate the oxidation of low-density lipoprotein and other lipoproteins which create proinflammatory lipid mediators that lead to a chronic inflammatory state. Over time, this chronic inflammatory state leads to plaque formation, its rupture, blood coagulation activation, thrombus formation and finally vessel occlusion [4,15]. Formation of F_2_-isoprostanes occurs in atherosclerotic lesions to a greater extent than in normal vessels, and their levels associated with the number of affected epicardial arteries, as shown in patients undergoing coronary angiography for suspected coronary artery disease [16,17].

In this analysis, only preoperative enhanced oxidative stress, assessed by 8-iso-PGF_2α_, measured in stable clinical condition, showed a correlation with long-term cardiovascular mortality. It is probable that oxidative stress that is enhanced by an acute large injury following surgery is better reflected by other than F_2_-isoprostanes pathways of pro- and/or antioxidant systems [4]. This hypothesis is supported by a significant increase of ADMA just after the operation and its association with worse cardiovascular long-term survival rate. Oxidative stress reduces the activity of dimethylarginine dimethylaminohydrolase, the enzyme that degrades ADMA, therefore leading to ADMA accumulation and as a result, inhibition of nitric oxide synthase. This process reduces the production of nitric oxide that is known be of key importance in endothelial dysfunction, the crucial step in atherogenesis [7]. Additionally, a genetically impaired availability of nitric oxide related to gene polymorphism of endothelial nitric oxide synthase may affect atherosclerotic plaque stability. It has been reported that the T-786C endothelial nitric oxide synthase genotype was associated with severe coronary artery disease and enhanced forearm vasodilatation indicating increased NO bioactivity. Genetically, enhanced nitric oxide production in the setting of oxidative stress can lead to poroxynitrate formation and thereby plaque destabilization [18]. Further studies are needed to assess occurrence of mentioned gene polymorphism among CABG population and its influence on mortality following surgery. Our previous report showed a positive correlation between 8-iso-PGF_2α_ and ADMA at all-time points, including pre- and postoperative measurements. Both markers increased significantly just after CABG and then markedly decreased 5–7 days following surgery, however, their concentrations were still greater than at baseline. Importantly, the increase of perioperative plasma ADMA cannot be explained by renal dysfunction, because renal failure was an exclusion criterion for this analysis [8]. The data suggest a relationship between oxidative stress and the severity of coronary atherosclerosis. Heslop et al. showed that another pro-oxidant enzyme—myeloperoxidase accurately predicted cardiovascular mortality risk in patients with coronary artery disease who were followed-up for more than 13 years [10]. Other studies suggest a positive association of circulating ADMA with cardiovascular long-term mortality in various (but not surgical) groups of patients [19,20]. Moreover, increased oxidative stress just after surgery was associated with all-cause of death. These findings could suggest that patients with greater long-term risk of death are more susceptible for all triggers (not only cardiovascular risk factors) inducing disturbances of the homeostatic state of enzymatic and non-enzymatic oxidant system [4]. Figure 1, which illustrates Kaplan–Meier curves showing probability of cardiovascular death in patients with increased baseline 8-iso-PGF_2α_, deserves a comment. In this figure both curves come together at 72 months and only diverge later on. It is possible that as people age, they become more sensitive to consequences of all causes that stimulate changes of the oxidant system, reflected in this study, by 8-iso-PGF_2α_.

Statins treatment was correlated with better cardiovascular survivals in the present study. Statins can influence the cardiovascular events by modulating the oxidative balance inhibiting synthesis of pro-oxidant compounds including 8-iso-PGF_2α_ [4]. Additionally, statins can affect ADMA metabolism. Serban et al. reported a significant reduction in plasma ADMA concentrations following statin therapy [21]. Addition of low-dose rivaroxaban to aspirin may further decrease morbidity and mortality rates in patients after CABG, however, this issue and the mechanisms involved require further study [22,23].

This study has several limitations. The size of the analyzed group was limited and confined to advanced coronary artery disease patients, however representative of CABG patients. No antioxidants were assessed in blood samples. Moreover, all laboratory variables were evaluated only at three time points. Windows of post-surgery measurements were large and a narrower measurement procedure might have led to lower variation of results, although our key finding was based on preoperative values unaffected by the surgery itself. Additionally, during follow-up, we recorded only one hard endpoint defined as death (cardiovascular or from other cause) or survival. We did not record major adverse cardiac and cerebrovascular events such as repeat revascularization, myocardial infarction and cerebrovascular accident or stroke. Moreover, many confounding factors could have occurred during the follow-up period which assessed only the vital status and cause of death. It is probable that knowledge of timing of adverse events during long-term follow-up could give better insight regarding the correlation between CABG and oxidative stress or ADMA. Additionally, lack of a control (non-CABG) group also limits the interpretation of the results. Therefore, the predictive power of 8-iso-PGF_2α_ and ADMA need to be assessed carefully. Further and larger studies could confirm our findings and provide evidence of exact mechanisms behind the reported associations. Still, the link between oxidative stress and long-term post CABG mortality is a novel observation which might have practical implications.

## 5. Conclusions

Our findings indicate that perturbation of the oxidative balance state generates consequences for long-term risk of death in patients after isolated, on-pump bypass surgery. Correlations between perioperative 8-iso-PGF_2α_ and ADMA and unfavorable long-term outcomes might suggest that these biomarkers could be useful in identifying high risk CABG patients.

## Figures and Tables

**Figure 1 jcm-11-00246-f001:**
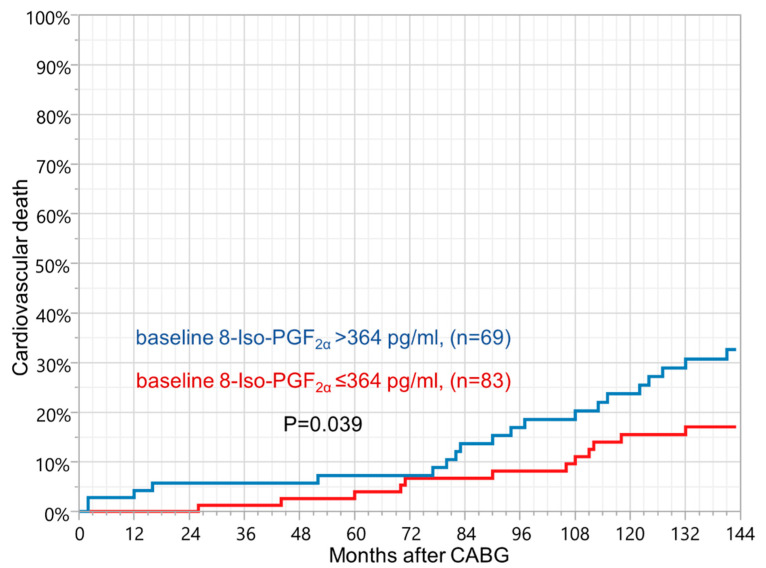
Kaplan–Meier curves showing probability of cardiovascular death in patients with increased baseline 8-iso-PGF_2α_ concentration.

**Figure 2 jcm-11-00246-f002:**
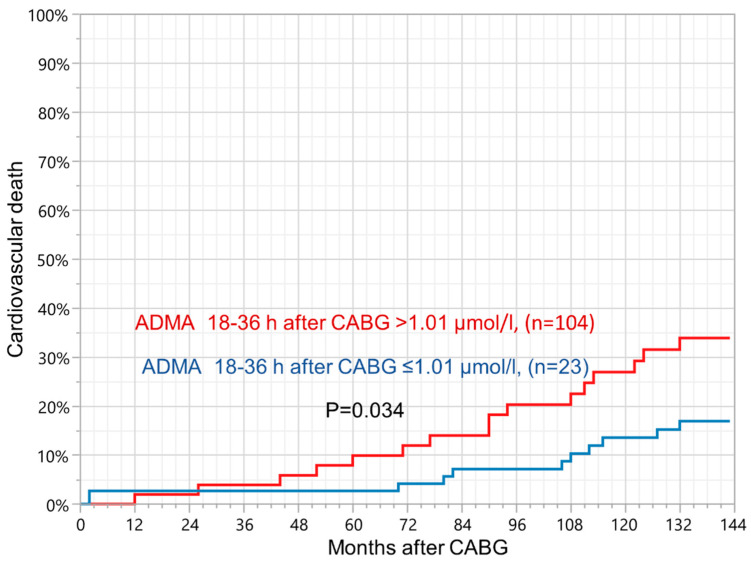
Kaplan–Meier curves showing probability of cardiovascular death in patients with increased ADMA concentration, 18–36 h after CABG.

**Table 1 jcm-11-00246-t001:** Perioperative characteristics of patients.

Variable at Time of CABG	All Patients (*n* = 152)	All Deaths			Cardiovascular Death		
		No (*n* = 84)	Yes (*n* = 68)	*p*-Value	No (*n* = 120)	Yes (*n* = 32)	*p*-Value
Age (years)	65.2 ± 8.3	63.0 ± 8.1	67.9 ± 7.9	**<0.001**	64.4 ± 7.8	67.9 ± 9.6	**0.037**
Male, *n* (%)	118 (77.6)	65 (77.4)	53 (77.9)	0.688	94 (78.3)	24 (75.0)	0.687
BMI (kg/m^2^)	28.0 ± 3.8	28.6 ± 3.7	27.3 ± 3.9	**0.049**	28.2 ± 3.9	27.2 ± 3.7	0.189
Peripheral vascular disease, *n* (%)	24 (15.8)	11 (13.1)	13 (19.1)	0.311	18 (15.0)	6 (18.7)	0.605
Type 2 diabetes, *n* (%)	43 (28.3)	18 (21.4)	25 (36.8)	**0.037**	31 (25.8)	12 (37.5)	0.193
Insulin, *n* (%)	22 (14.5)	10 (11.9)	12 (17.6)	0.317	17 (14.2)	5 (15.6)	0.783
Hypertension, *n* (%)	123 (80.9)	69 (82.1)	54 (79.4)	0.670	100 (83.3)	23 (71.9)	0.143
Preoperative MI, *n* (%)	123 (80.9)	67 (79.8)	56 (82.3)	0.686	97 (80.8)	26 (81.2)	0.957
Postoperative MI, *n* (%)	13 (8.5)	6 (7.1)	7 (10.3)	0.410	10 (8.3)	3 (9.4)	1.000
Dyslipidemia, *n* (%)	98 (64.5)	58 (69.0)	40 (58.8)	0.190	81 (67.5)	17 (53.1)	0.131
Previous PCI, *n* (%)	18 (11.8)	11 (13.1)	7 (10.3)	0.595	15 (12.5)	3 (9.4)	0.766
COPD, *n* (%)	7 (4.61)	4 (4.8)	3 (4.4)	1.000	5 (4.2)	2 (6.2)	0.638
ACE inhibitors, *n* (%)	133 (87.5)	74 (88.1)	59 (86.8)	0.805	105 (87.5)	28 (87.5)	1.000
Beta blockers, *n* (%)	135 (88.8)	79 (94.0)	56 (82.3)	**0.023**	107 (89.2)	28 (87.5)	0.758
Statins, *n* (%)	135 (88.8)	77 (91.7)	58 (85.3)	0.215	110 (91.7)	25 (78.1)	**0.047**
EuroSCORE I (points)	2.9 ± 1.8	2.5 ± 1.7	3.5 ± 1.8	**<0.001**	2.7 ± 1.8	3.8 ± 1.9	**0.002**
ICU length of stay ≥2 days, *n* (%)	58 (38.2)	26 (30.9)	32 (47.1)	**0.042**	40 (33.3)	18 (56.2)	**0.018**

ACE: angiotensin-converting enzyme; BMI: body mass index; CABG: coronary artery bypass grafting; COPD: chronic obstructive pulmonary disease; ICU: intensive care unit; MI: myocardial infarction; PCI: percutaneous coronary interventions. Values are displayed as mean ± standard deviation or number (percentage). Bold values denote statistical significance at the *p* < 0.050 level.

**Table 2 jcm-11-00246-t002:** The causes of death according to the International Classification of Diseases (ICD-10).

Cause of Death (ICD-10 Codes)	Number of Patients (%)
Cardiovascular diseases	32 (47.06)
Heart diseases (I10–I52)	22 (32.35)
Cerebrovascular diseases (I60–I69)	4 (5.88)
Other diseases of the circulatory system (I70–I99)	6 (8.82)
Malignant neoplasms (C00–C97)	16 (23.53)
Respiratory diseases (J00–J99)	11 (16.18)
Accidents (V01–X59)	2 (2.94)
Diabetes mellitus (E10–E14)	1 (1.47)
Other diagnosis	6 (8.82)
Age-related physical debility (R54)	1 (1.47)
Ill-defined and unknown cause of mortality (R99)	1 (1.47)
Parkinson’s disease (G20)	1 (1.47)
Other bacterial diseases, not elsewhere classified (A48)	1 (1.47)
Chronic kidney disease (N18)	1 (1.47)
Neoplasm of uncertain behavior of brain and central nervous system (D43)	1 (1.47)

**Table 3 jcm-11-00246-t003:** The impact of clinical variables on hazard ratios for death during follow-up after adjustment for age and sex.

Variable at Time of CABG	All Deaths		Cardiovascular Death	
	Hazard Ratio (95% CI)	*p*-Value	Hazard Ratio (95% CI)	*p*-Value
BMI	0.936 (0.874–1.000)	**0.049**	0.926 (0.837–1.020)	0.123
Peripheral vascular disease	1.280 (0.692–2.365)	0.442	1.301 (0.528–3.205)	0.577
Type 2 diabetes	1.643 (1.002–2.696)	**0.048**	1.645 (0.802–3.376)	0.185
Insulin use	1.404 (0.753–2.620)	0.286	1.187 (0.457–3.084)	0.724
Hypertension	0.972 (0.532–1.775)	0.926	0.606 (0.273–1.347)	0.235
Preoperative MI	0.918 (0.491–1.718)	0.791	0.826 (0.338–2.017)	0.680
Postoperative MI	1.685 (0.756–3.755)	0.230	1.444 (0.429–4.859)	0.570
Dyslipidemia	0.962 (0.583–1.587)	0.879	0.755 (0.368–1.550)	0.445
Previous PCI	0.765 (0.350–1.673)	0.502	0.684 (0.208–2.245)	0.531
COPD	0.984 (0.309–3.131)	0.978	1.451 (0.347–6.077)	0.610
ACE inhibitors	0.919 (0.456–1.853)	0.813	0.970 (0.340–2.766)	0.955
Beta blockers	0.493 (0.264–0.920)	**0.026**	0.711 (0.249–2.028)	0.523
Statins	0.649 (0.331–1.270)	0.207	0.397 (0.171–0.919)	**0.031**
EuroSCORE I	1.183 (1.003–1.383)	**0.046**	1.368 (1.086–1.703)	**0.008**
ICU length of stay ≥ 2 days	1.704 (1.058–2.744)	**0.028**	2.477 (1.232–4.983)	**0.011**

ACE: angiotensin-converting enzyme; BMI: body mass index; CABG: coronary artery bypass grafting; CI: confidence interval; COPD: chronic obstructive pulmonary disease; ICU: intensive care unit; MI: myocardial infarction; PCI: percutaneous coronary interventions. Bold values denote statistical significance at the *p* < 0.050 level.

**Table 4 jcm-11-00246-t004:** Plasma concentrations of 8-iso-prostaglandin F_2α_ and asymmetric dimethylarginine.

Variable at Time of CABG	All Patients (*n* = 152)	All Deaths			Cardiovascular Death		
		No (*n* = 84)	Yes (*n* = 68)	*p*-Value	No (*n* = 120)	Yes (*n* = 32)	*p*-Value
8-iso-PGF_2α_							
baseline (pg/mL)	357 ± 38.4	353 ± 36.2	362 ± 40.5	0.132	354 ± 38.7	367 ± 35.7	**0.045**
baseline, *n* (%) (ROC optimal cut off >364 pg/mL for cardiovascular death)	81 (53.3)	49 (58.3)	32 (47.0)	0.166	69 (57.5)	12 (37.5)	**0.044**
18–36 h after CABG (pg/mL)	465 ± 40.1	459 ± 35.7	473 ± 43.7	**0.032**	465 ± 40.4	464 ± 39.7	0.892
difference between baseline and 18–36 h after CABG, *n* (%) (ROC optimal cut off >135 pg/mL for cardiovascular death)	89 (58.5)	48 (57.1)	41 (60.3)	0.681	66 (55.0)	23 (71.9)	**0.042**
5–7 days after CABG (pg/mL)	414 ± 43.4	410 ± 32.9	417 ± 51.4	0.935	413 ± 40.5	416 ± 53.8	0.803
ADMA							
baseline (μmol/L)	0.56 ± 0.06	0.56 ± 0.05	0.56 ± 0.07	0.433	0.56 ± 0.06	0.57 ± 0.06	0.164
18–36 h after CABG (μmol/L)	0.93 ± 0.10	0.93 ± 0.08	0.94 ± 0.12	0.780	0.94 ± 0.09	0.92 ± 0.11	0.111
18–36 h after CABG, *n* (%) (ROC optimal cut off >1.01 μmol/L for cardiovascular death)	52 (34.2)	26 (30.1)	26 (38.2)	0.334	36 (30.0)	16 (50.0)	**0.029**
difference between baseline and 18–36 h after CABG, *n* (%) (ROC optimal cut off >0.44 μmol/L for cardiovascular death)	44 (28.9)	23 (27.4)	21 (30.9)	0.639	30 (25.0)	14 (43.7)	**0.034**
5–7 days after CABG (μmol/L)	0.74 ± 0.11	0.73 ± 0.09	0.75 ± 0.12	0.898	0.74 ± 0.10	0.74 ± 0.13	0.829

8-iso-PGF_2α_: 8-iso-prostaglandin F_2α_; ADMA: asymmetric dimethylarginine; CABG: coronary artery bypass grafting. Values are shown as mean ± standard deviation or number (percentage). Bold values denote statistical significance at the *p* < 0.050 level.

**Table 5 jcm-11-00246-t005:** 8-iso-prostaglandin F_2α_ and asymmetric dimethylarginine hazard ratios for death during follow-up (adjusted for age and sex).

Variable at Time of CABG	All Deaths		Cardiovascular Death	
	Hazard Ratio (95% CI)	*p*-Value	Hazard Ratio (95% CI)	*p*-Value
8-iso-PGF_2α_				
Baseline	1.006 (1.000–1.013)	0.058	1.010 (1.001–1.021)	**0.036**
Baseline (ROC optimal cut off ≤364 pg/mL for lower risk of cardiovascular death)	0.665 (0.412–1.072)	0.093	0.460 (0.224–0.942)	**0.030**
18–36 h after CABG	1.007 (1.000–1.014)	**0.048**	1.000 (0.990–1.010)	0.973
difference between baseline and 18–36 h after CABG (ROC optimal cut off >135 pg/mL for cardiovascular death)	1.222 (0.685–2.182)	0.491	2.701 (0.932–7.829)	**0.041**
5–7 days after CABG	1.002 (0.994–1.010)	0.604	1.000 (0.988–1.012)	0.947
ADMA				
baseline	13.589 (0.131–1505.058)	0.272	350.694 (0.358–377,546.745)	0.095
18–36 h after CABG	4.544 (0.199–98.162)	0.340	0.147 (0.001–14.194)	0.415
18–36 h after CABG (ROC optimal cut off >1.01 μmol/L for cardiovascular death)	1.417 (0.839–2.394)	0.195	2.467 (1.140–5.340)	**0.020**
difference between baseline and 18–36 h after CABG (ROC optimal cut off >0.44 μmol/L for cardiovascular death)	1.186 (0.685–2.053)	0.545	2.192 (1.017–4.728)	**0.047**
5–7 days after CABG	6.681 (0.223–197.455)	0.274	2.079 (0.011–375.091)	0.785

8-iso-PGF_2α_: 8-iso-prostaglandin F_2α_; ADMA: asymmetric dimethylarginine; CABG: coronary artery bypass grafting; CI: confidence interval. Bold values denote statistical significance at the *p* < 0.050 level.

## Data Availability

The data presented in this study are available on request from the corresponding author.

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
