# Peer review of "8-Isoprostanes and Asymmetric Dimethylarginine as Predictors of Mortality in Patients Following Coronary Bypass Surgery: A Long-Term Follow-Up Study"

_jcm, 2022, doi:10.3390/jcm11010246_

Round 1

Reviewer 1 Report

  1. The long-term predictive value of 8-iso-prostaglandin F2a and asymmetric dimethylarginine was evidently confounded by multiple factors. Lack of a control (non-CABG) group significantly limited the interpretation of the results.
  2. As shown in Table 4, the values of 8-iso-PGF2a and ADMA showed minimal differences between the groups. The cutoff values used for Kaplan-Meier curves (Fig. 1 and Fig.2) were somewhat arbitrary. Therefore, a control group becomes extremely important.
  3. While the sample size was very small with a total of only 152 cases followed for 12 years, the key determinant of survival/death appeared to be the age as showed in the Table 1. For example, at the time of surgery, survivor group averaged 63.0 yr, as compared with those who died were 67.9 yr. The patients were followed up for 12 years. Of note, the life expectancy in Poland is 77 yr.  In this case, the predictive power of  8-iso-PGF2a and ADMA is questionable.

Author Response

Dear Reviewer of International Journal of Clinical Medicine,

Thank you very much for your patience and all the valuable comments and suggestions. I tried my best to address all your concerns. Please find my answers below.

Sincerely yours,

Dariusz Plicner on behalf of the authors.

  1. The long-term predictive value of 8-iso-prostaglandin F2a and asymmetric dimethylarginine was evidently confounded by multiple factors. Lack of a control (non-CABG) group significantly limited the interpretation of the results.

Thank you for your valuable comment. Because this is observational and long-term retrospective study, we are unable to match the control group. We have enhanced the information in Limitations section, according to your suggestion (page 19, line 304).

  1. As shown in Table 4, the values of 8-iso-PGF2a and ADMA showed minimal differences between the groups. The cutoff values used for Kaplan-Meier curves (Fig. 1 and Fig.2) were somewhat arbitrary. Therefore, a control group becomes extremely important.

Yes, it is right. Therefore, we emphasized limitations of our research in the Limitations section (page 19, line 305).

  1. While the sample size was very small with a total of only 152 cases followed for 12 years, the key determinant of survival/death appeared to be the age as showed in the Table 1. For example, at the time of surgery, survivor group averaged 63.0 yr, as compared with those who died were 67.9 yr. The patients were followed up for 12 years. Of note, the life expectancy in Poland is 77 yr. In this case, the predictive power of  8-iso-PGF2a and ADMA is questionable.

Thank you for your comment. Yes, there are several limitations of the research design. We emphasized these limitations of the study in the Limitations section, according to your suggestion (page 18-19, lines 291-293 and 304-305).

Reviewer 2 Report

The authors should be commended fo undertaking this long-term study. The results are novel and interesting and the several limitations of this study were acknowledged. The manuscript was written and assembled reasonably well

Unfortunately several important issues were disregarded and need authors attention.

  1. It has been reported in a number of study that genetically enhanced NO production in the setting of oxidative stress can lead to poroxynitrate formation and thereby plaque destabilization (JACC doi: 10.1016/s0735-1097(02)03012-7.). The authors should therefore discuss their findings in the light of this.
  2. The increased risk of iso-PGF2α was 1%,i.e. tiny ( 1.010 (1.001-1.021). It must be reported as such throughout the manuscript.
  3. Moreover, since the Authors rightly used Cox' regression, it is not clear why the entered in their multivariate prediction model only age and sex. As clearly there were predictors of CV death stranger than  iso-PGF2α and ADMA, as for example statins use, hypertension and diabetes, the key question that remains unsettled is if the two biomarkers will retain the prognostic role after this adjustments. This must be clarified.
  4. In fact, clinically the main question would be: does measurement of isoprostanes and ADMA convey prognostic information over what already known from the other well established prognosticators?
  5. To my knowledge, the Akaike information criterion (AIC) that they presented as 'stopping rule' is a mathematical method for evaluating how well a model fits the data it was generated from, as compared to other models.
  6. From what written it is totally unclear what the Authors did and this should be specified also to allow verification of the data.
  7. The latter should be made publicly available for incorporation by others in meta-analysis.

Author Response

Dear Reviewer of International Journal of Clinical Medicine,

Thank you very much for your patience and all the valuable comments and suggestions. I tried my best to address all your concerns. Please find my answers below.

Sincerely yours,

Dariusz Plicner on behalf of the authors.

  1. It has been reported in a number of study that genetically enhanced NO production in the setting of oxidative stress can lead to poroxynitrate formation and thereby plaque destabilization (JACC doi: 10.1016/s0735-1097(02)03012-7.). The authors should therefore discuss their findings in the light of this.

Based on mentioned article, we have described role of altered bioavailability of nitric oxide in atherosclerosis in the Discussion section (page 17, lines 260-263).

  1. The increased risk of iso-PGF2α was 1%,i.e. tiny ( 1.010 (1.001-1.021). It must be reported as such throughout the manuscript.

This increased risk was 1% per unit of 8-iso-PGF2α. We have changed „per unit” for „1 pg/ml” throughout in manuscript.

  1. Moreover, since the Authors rightly used Cox' regression, it is not clear why the entered in their multivariate prediction model only age and sex. As clearly there were predictors of CV death stranger than iso-PGF2α and ADMA, as for example statins use, hypertension and diabetes, the key question that remains unsettled is if the two biomarkers will retain the prognostic role after this adjustments. This must be clarified.

 Thank you for your valuable comment. We did not performed multivariate analysis adjusted for statins use or hypertension, because most of patients (about 90%) used statins and have had hypertension.

  1. In fact, clinically the main question would be: does measurement of isoprostanes and ADMA convey prognostic information over what already known from the other well established prognosticators?

Thank you for your comment. Yes, there are several limitations of the research design. We emphasized these limitations of the study in the Limitations section, according to your suggestion (page 18-19, lines 291-293 and 304-305).  

  1. To my knowledge, the Akaike information criterion (AIC) that they presented as 'stopping rule' is a mathematical method for evaluating how well a model fits the data it was generated from, as compared to other models. From what written it is totally unclear what the Authors did and this should be specified also to allow verification of the data.

Thank you for your comment. For clarity and better reading purposes, we deleted AIC information (page 6-7, lines127-128 and page 15, lines 221).

  1. The latter should be made publicly available for incorporation by others in meta-analysis.

Yes, naturally. We will send to the Editor the information regarding data availability statement: “The data that support the findings of this study are available upon request from the corresponding author [D.P.]“.

Round 2

Reviewer 1 Report

The authors addressed all questions raised by the Review, as the limitations of the study. 

Author Response

Dear Reviewer of International Journal of Clinical Medicine,

Thank you very much for your patience and all the valuable comments and suggestions. I tried my best to address all your concerns. Please find my answers below.

Sincerely yours,

Dariusz Plicner on behalf of the authors.

Comments and Suggestions for Authors:

The authors addressed all questions raised by the Review, as the limitations of the study.

            Thank you.

Reviewer 2 Report

The Authors have responded to my comments "It has been reported in a number of study that genetically enhanced NO production in the setting of oxidative stress can lead to poroxynitrate formation and thereby plaque destabilization " in a suboptimal way as they failed to mention that the T-786C SNP was associated with enhanced  forearm vasodilation indicating increased NO bioactivity. Yet, this gene variants was associated with a worse prognosis (JACC doi: 10.1016/s0735-1097(02)03012-7.). Therefore, they should better discuss their findings in light of this authors should therefore discuss their findings in the light of this and not as they did in the Discussion section (page 17, lines 260-263).

Author Response

Dear Reviewer of International Journal of Clinical Medicine,

Thank you very much for your patience and all the valuable comments and suggestions. I tried my best to address all your concerns. Please find my answers below.

Sincerely yours,

Dariusz Plicner on behalf of the authors.

Comments and Suggestions for Authors

The Authors have responded to my comments "It has been reported in a number of study that genetically enhanced NO production in the setting of oxidative stress can lead to poroxynitrate formation and thereby plaque destabilization " in a suboptimal way as they failed to mention that the T-786C SNP was associated with enhanced  forearm vasodilation indicating increased NO bioactivity. Yet, this gene variants was associated with a worse prognosis (JACC doi: 10.1016/s0735-1097(02)03012-7.). Therefore, they should better discuss their findings in light of this authors should therefore discuss their findings in the light of this and not as they did in the Discussion section (page 17, lines 260-263).

            Thank you for your comment. We discussed above findings more precisely. We hope that in better way than before (page 17, lines 258-267).